# Improving Chemical Reaction Prediction with Unlabeled Data

**DOI:** 10.3390/molecules27185967

**Published:** 2022-09-14

**Authors:** Yu Xie, Yuyang Zhang, Ka-Chun Wong, Meixia Shi, Chengbin Peng

**Affiliations:** 1College of Information Science and Engineering, Ningbo University, Ningbo 315211, China; 2Department of Computer Science, City University of Hong Kong, Hongkong 999077, China; 3College of Chemical Engineering, Ningbo Polytechnic, Ningbo 315000, China

**Keywords:** chemical reaction prediction, semi-supervised learning, Mean Teacher Weisfeiler–Lehman Network

## Abstract

Predicting products of organic chemical reactions is useful in chemical sciences, especially when one or more reactants are new organics. However, the performance of traditional learning models heavily relies on high-quality labeled data. In this work, to utilize unlabeled data for better prediction performance, we propose a method that combines semi-supervised learning with graph convolutional neural networks for chemical reaction prediction. First, we propose a Mean Teacher Weisfeiler–Lehman Network to find the reaction centers. Then, we construct the candidate product set. Finally, we use an Improved Weisfeiler–Lehman Difference Network to rank candidate products. Experimental results demonstrate that, with 400k labeled data, our framework can improve the top-5 accuracy by 0.7% using 35k unlabeled data. When the proportion of unlabeled data increases, the performance gain can be larger. For example, with 80k labeled data and 35k unlabeled data, the performance gain with our framework can be 1.8%.

## 1. Introduction

The use of machine learning methods can help researchers to make progresses in various fields. They have shown great potential in chemistry, including quantum chemistry, density functional analysis, drug design, reaction prediction, and retrosynthesis analysis [1]. Some researchers used machine learning to carry out retrosynthesis research on 12 anti-COVID-19 drugs that are still in the research stage [2], trying to use cheap and readily available raw materials for drug synthesis. Staszak et al. [3] used machine learning to explore the relationship between chemical structure–biological activity, etc. The prediction accuracy of the current mainstream single-step retrosynthesis reaction models still has the potential to improve. For example, the top-1 accuracy through the machine translation model [4] on the USPTO test set is 58.3%. When it is difficult to improve the performance of the single-step retrosynthesis reaction model, it is particularly critical to use a set of reliable forward organic chemical reaction prediction algorithms to screen out the unreasonable reactions recommended in the single-step retrosynthesis reaction model. In this research work, we focus on the application of chemical reaction prediction.

Chemical reaction prediction has the purpose to predict the corresponding products with given reactants, reagents, solvents, etc. Chemical reaction prediction is one of the key steps in the preparation of organic materials; however, predicting organic compounds and their retrosynthetic analysis with high accuracy and efficiency is still a challenging problem.

Currently, methods for chemical reaction prediction can be classified into three categories: rule-based expert systems, quantum mechanical simulations, and machine learning-based systems. Several computer assistance synthetic design [5,6,7] systems have appeared since the 1960s, such as Logic and Heuristics Applied to Synthetic Analysis (LHASA) [8]. However, traditional methods use rules formulated by a large number of experts to judge the feasibility of a certain path, and do not achieve completely satisfactory results [9]. First-principles-based quantum mechanical simulations can obtain accurate prediction results [10], but the results of such methods are closely related to theoretical models and calculation parameters.

With the enrichment of chemical reaction data entries, combined with recently developed deep learning methods, researchers have developed product prediction methods based on chemical reaction templates. For example, Wei et al. [11] were the first to conceptually demonstrate the feasibility of deep learning to predict reaction products. For a given reactant and reagent, 16 similar chemical reaction templates are generated by using the simulation data, so as to deduce the corresponding products. Segler et al. [12,13] used experimental data and extended this method, using nearly 10,000 templates generated by the algorithm to deduce the probability distribution of the product; then, the product was evaluated, and the compound with the highest evaluation score was regarded as the main product.

Although the above template-based reaction prediction models can predict the main products with high accuracy, the products are limited to the predicted range of known templates. This limits the possibility of machine learning models predicting new products. To overcome such limitation, template-free chemical reaction prediction models are proposed. One is the Sequence-to-Sequence (Seq2sep) model [14,15]. The idea of this model for chemical reaction prediction is to convert the reactants and reagents represented in the simplified molecular-input line-entry system (SIMILES) to products in SIMILES as well. This model consists of two distinct recurrent neural networks (RNN) and integrates the attention mechanism, which is useful for predicting atom-to-atom mapping. Bort et al. [16] first attempted to use a combination of condensed reaction maps, generating topological maps and sequence-to-sequence autoencoders to generate new chemical reactions.

Using graph convolutional neural networks to predict atom and chemical bond changes is also a machine learning strategy for reaction prediction. Coley et al. [17] used a molecular graph to represent reactant molecules. The probabilities of chemical bond changes between each atom pair were calculated by a graph convolutional neural network, and candidate products were enumerated in combination and the probability distribution of the main products was re-predicted by another graph convolutional network.

The accuracy of the predicted products of the above methods heavily relies on the labeled datasets. However, obtaining a large amount of labeled data is expensive. To address the difficulty of data acquisition, Hao et al. [18] used an active semi-supervised graph convolutional neural network to predict molecule properties by combining labeled and unlabeled data. Chen et al. [19] used a combination of mean teacher and graph convolutional neural networks for chemical toxicity prediction. Schwaller et al. [20] used a transformer network to learn atomic mappings between reactants and products of chemical reactions without any human labeling or supervision. These methods demonstrate that unlabeled data can also positively impact model performance.

To reduce the demand for fully labeled data while keeping the performance of prediction algorithms, we propose to use a semi-supervised framework for template-free graph convolutional neural networks to predict chemical reactions. First, we convert the reactants to a graph, and then apply some perturbations to the reactant graph. The perturbed graph is input into the student model, and the unperturbed graph is input into the teacher model. Next, we iteratively update the parameters of the student and teacher models to get the reaction centers. A set of candidate products is obtained by enumerating all cases in the set of reaction centers. Finally, these candidate products are input into another graph convolutional neural network to obtain a probability distribution of the candidate products, and then find the final products.

The contributions of this work is as follows.
We propose to integrate chemical product prediction models with a mean-teacher framework to improve the performance in predicting unknown chemical reactions.We propose a novel perturbation method with random atom feature removal for the framework.Experiments demonstrate the effectiveness of our approach that, with the help of unlabeled data, further improves the prediction accuracy.

We use the United States Patent and Trademark Office (USPTO) dataset [21] and the Schneider dataset [22] to validate our model. To the best of our knowledge, this is the first work in which a semi-supervised approach has been used to predict chemical reactions.

## 2. Experiments

### 2.1. Datasets

In these experiments, we use the USPTO dataset as the labeled dataset and the Schneider dataset as the unlabeled dataset with reaction products removed. Although our approach is applicable on many chemical reaction datasets, in this experiment, we choose the popular USPTO dataset for the ease of comparison with other approaches [15,17,23]. We apply the default split of the USPTO dataset, which divides the dataset into three parts, 400 k/40 k/30 k, as the train set/test set/validation set, respectively. The default split for the training set, test set and validation set has the same proportion as reaction of various types. We checked that the Schneider dataset and the USPTO dataset do not have the same items, so we use Schneider as the unlabeled dataset. We use tensorflow2.4 and rdkit for python in our research.

### 2.2. Find Reaction Center

The task of this model is to output a set of reaction centers. The label is also a set of reaction centers. Let TR(p) be the true reaction center set, R(p) be the reaction center predicted by the model. If R(p) contains all elements in TR(p), we consider this is a correct prediction. We set hyperparameters for this model with depth = 3, the initial learning rate is 0.01, and it is reduced by 5% for every 10,000 training steps. With the help of unlabeled data, the accuracy of finding reaction centers is 87.3% higher than for the WLN+I-WLDN model for which it is 87.0%. This means that the unlabeled data can improve the accuracy of the model.

After predicting the reaction centers, we want to generate a set of candidate products. We analyzed the dataset as shown in Figure 1. Because most of the reactions contain five or fewer reaction centers, we choose the top-5 reaction centers to construct candidate products. The number of candidate products is therefore ∑n=155n, where .. is the binomial coefficient. Therefore, there are only 31 candidate products for each reactant.

### 2.3. Candidate Products Ranking

The final step is to select the true product from the candidate product set. We choose some reaction centers, and then construct products based on the reactants and reaction centers. We compare these products with the real product, and if any of them can match, then one prediction is correct. Next, we compare our work (MT-WLN+I-WLDN) with other reaction prediction methods and the comparison results are shown in Table 1. We improved the top-5 accuracy by 0.7% with the help of unlabeled datasets. We also found that the accuracy of the model increases as the unlabeled data increase. With the increase of unlabeled data, the accuracy of the model continues to rise, which also shows that the unlabeled data can improve the performance of the model.

We randomly select some of the labeled data for training, to see the performance of our framework. We choose the WLN+Improved WLDN model as the benchmark approach. We use 20%, 50%, 70%, and 100% labeled data, respectively, from the USPTO dataset and all the unlabeled data from the Schneider 35k dataset to train the models. The test results are shown in Figure 2. We find that with fewer labeled data, the accuracy of our model decreases less than the WLN+I-WLDN model, and the accuracy of our model is 1.8% higher than WLN+I-WLDN at 20% labeled data. This shows that in the case of fewer labeled data, unlabeled data play a greater role. We take the reaction of propylene oxide and pyrroloquinoline as an example to explain our model. Figure 3 shows the attention score of the reaction center. The darker the color of the atom, the higher the attention score. The reaction center atom is marked in green and the attention score is marked in blue. The attention score can reflect the connection of chemical bonds. In the product, the atom 14 C is connected to the atom 1 N; so, this approach makes atom 14 C allocate more attention to the atom 1 N. However, this approach can also assign a smaller attention to Atom 3 C and Atom 10 N, which may result in by-products.

## 3. Method

In this paper, we propose a novel Mean Teacher Weisfeiler–Lehman Network (MT-WLN) for organic reaction prediction by using both labeled and unlabeled reactions. The flowchart of the whole approach is depicted in Figure 4.

For the first step, we iteratively used the teacher and student models. Each of them is a Weisfeiler–Lehman Network (WLN) [24]. In the student network, we use the perturbed WLN to find the atom pairs most likely to react. In the teacher network, we use the Exponential Moving Average (EMA) of the student model to update the parameters. Then, we use the teacher model to assign pseudo-labels for the unlabeled dataset. This way, the student model can learn from these pseudo-labels. After that, we apply the attention mechanism with the goal of capturing the fact that atoms outside of the reaction center may be necessary for the reaction to occur. For the second step, we select the top-K atom pairs with the highest predicted reactivity scores. Then, we enumerate all possible bond configuration changes in the set to obtain the set of candidate products. Finally, we use the Improved Weisfeiler–Lehman Difference Network (I-WLDN) [23] to rank candidate products, so that the main product has the highest score.

### 3.1. Find Reaction Center

We represent a molecule as a graph, atoms as nodes, and bonds as edges. A chemical reaction is represented as a transformation from graph $G_{r}$ to graph $G_{p}$, where $G_{r}$ are the reactants and $G_{p}$ is the product. Both reactants and product are atom-mapped so that we can easily find the changing bonds and then find the reaction center. The reaction center is represented by a tuple ($atom_{1}$, $atom_{2}$, $bond_{new}$), in which the bond between $atom_{1}$ and $atom_{2}$ can be changed into one of the four types: no bond, single bond, double bond, and aromatic bond. The MT-WLN workflow is depicted in Figure 5.

First, we need to perturb the input data. The input data include atom features $f_{a}$ (including atomic number, mass, aromaticity, connectivity, and valence), bond features $f_{b}$ (including bond type, whether conjugated, whether cyclic), reactant graph $G_{r}$ (including an adjacency list, representing neighbors of each atom), and a reactant bond graph $G_{rb}$ (including an adjacency list representing the bonds each atom is connected to). We input the perturbed data to the student model to allow the student model to be resistant to noise, and then learn more from the teacher model to enhance robustness. We define the perturbation function as $p(G_{r})$, where each atom has probability λ of discarding that atom’s features. The task of the teacher model is to construct better labels for student model to learn. The network structure of the teacher model is the same as the student model. The differences between the student and the teacher are in the input data and the parameter update method. We choose unperturbed data as the input to the teacher network. Then, the set of reaction centers is obtained by the following formula:(1)$$R{\left(p\right)}_{stu}=WLN_{stu}\left(W_{stu},f_{a},f_{b},p\left(G_{r}\right),G_{rb}\right),$$
(2)$$R{\left(p\right)}_{tea}=WLN_{tea}\left(W_{tea},f_{a},f_{b},G_{r},G_{rb}\right),$$
where WLN(·) is a prediction function based on Weisfeiler–Lehman Network [24], R(p) is a set containing predicted reaction centers, *W* is the weight matrix. Then, for the labeled data, we compute the cross-entropy loss:(3)$$L_{CEL}=-\sum_{s_{i}\in R\left(p\right)}y_{i}log\left(s_{i}\right)+\left(1-y_{i}\right)log\left(1-s_{i}\right),$$
where $s_{i}\in R\left(p\right)$ is the *i*th reaction center predicted by student model. The cross-entropy loss is a metric used to measure how a classification algorithm performs in machine learning. The predicted class is compared with the ground truth class, and the cross-entropy loss is computed. It is between 0 and 1, with 0 indicating a good prediction. Our goal is generally to optimize the algorithm and minimize the cross-entropy loss. For both labeled and unlabeled data, we train to minimize the mean-squared error between the teacher model and the student model:(4)$$L_{MSE}=\frac{1}{n}\sum_{\begin{array}{c} s_{tea}^{i}\in R{\left(p\right)}_{tea}\\ s_{stu}^{i}\in R{\left(p\right)}_{stu} \end{array}}^{n}{\left(s_{tea}^{i}-s_{stu}^{i}\right)}^{2}.$$

The mean-squared error (MSE) is defined as the average of the squares of the differences between the ground-truth and the predicted values. The closer the value of MSE is to 0, the closer the prediction of the student model is to the prediction of the teacher model. Then, we combine the two losses to update the weight matrix of the student model:(5)$$L_{stu}=L_{CEL}+\omega L_{MSE},$$
where ω can determine the weight of $L_{MSE}$ in the total loss. After updating the parameters of the student model, we use the Exponential Moving Average of the student model to update the weight matrix of the teacher model:(6)$$\theta_{t}^{i}=\alpha \theta_{s}^{i-1}+\left(1-\alpha \right)\theta_{s}^{i},$$
where $\theta_{s}^{i-1}\in W_{stu}^{i-1}$ is the student parameter for the (i−1)th iteration, $\theta_{s}^{i}\in W_{stu}^{i}$ is the student parameter for the (i)th iteration, and $\theta_{t}^{i}\in W_{tea}^{i}$ is the teacher parameter for the (i)th iteration.

### 3.2. Candidate Products Ranking

We select the top-*k* atom pairs with the highest reaction scores form the reaction center set, and then enumerate each case in the set to form the candidate product set. The score of every candidate product can be obtained by:(7)$$s\left(p_{i}\right)=WLDN\left(W,rc\left(p_{i}\right),f_{a},f_{b},G_{r},G_{rb}\right),$$
where WLDN(·) is the Weisfeiler–Lehman Difference Network [23], $rc(p_{i}$) is the set of reaction centers from reactant $r_{i}$ to product $p_{i}$, $s(p_{i})$ is the product $p_{i}$’s score. Then, we train the model to minimize the softmax log-likelihood objective over scores $\left\{s\left(p_{0}\right),s\left(p_{1}\right),\ldots ,s\left(p_{m}\right)\right\}$, where $s(p_{0})$ is the true product’s score.

## 4. Conclusions

In this paper, we propose a framework that enables unlabeled data to be used for training chemical reaction prediction algorithms which performs better than traditional approaches using labeled data only. Experimental results verify that the unlabeled data are not useless and the unlabeled data can enhance model performance. Nevertheless, our model still has some limitations. For example, the graph convolutional network obtains local and global information by iteratively gathering neighborhood information, which may lead to over-smoothing, namely, all the nodes sharing similar features. In this work, we do not consider the three-dimensional structure of chemical molecules either. Our framework can improve the performance of traditional methods in terms of prediction accuracy with the same amount of labeled data. We believe that this framework can help chemists in discovering new chemical reactions in the future.

## Figures and Tables

**Figure 1 molecules-27-05967-f001:**
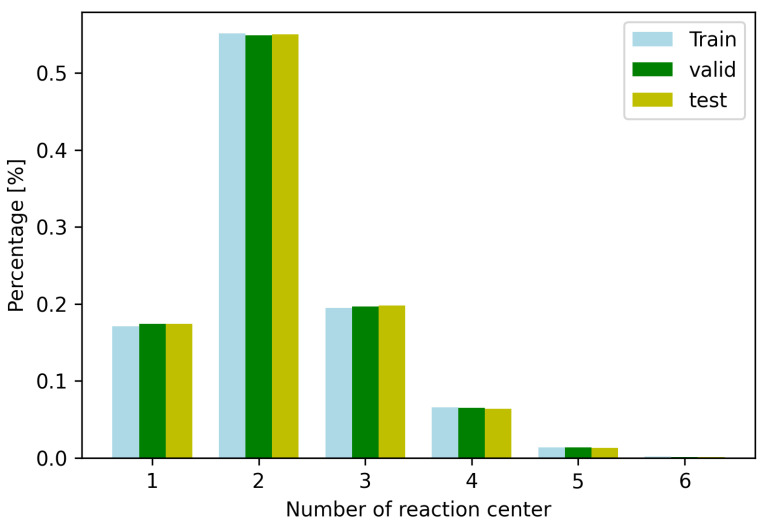
The number of reaction centers gained from the USPTO train/test/validation datasets. There are only few reactions which include 6 reaction centers.

**Figure 2 molecules-27-05967-f002:**
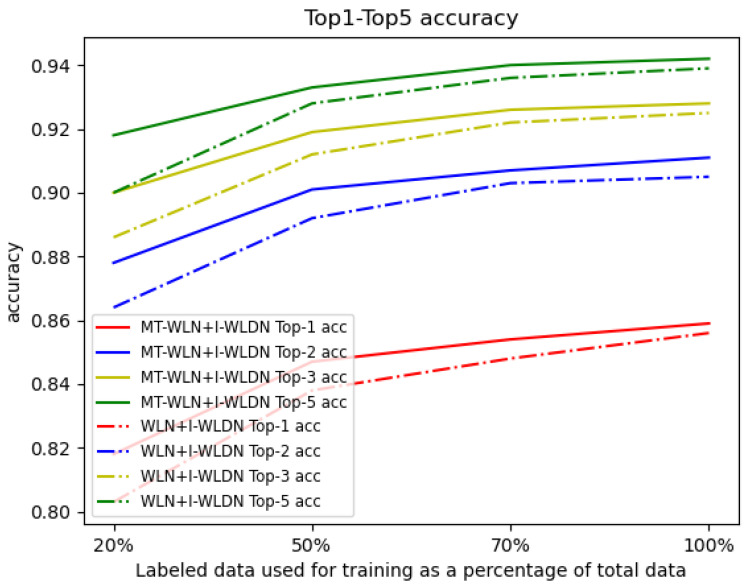
The top-1, top-2, top-3, and top-5 accuracies with labeled data size from 20% to 100%.

**Figure 3 molecules-27-05967-f003:**
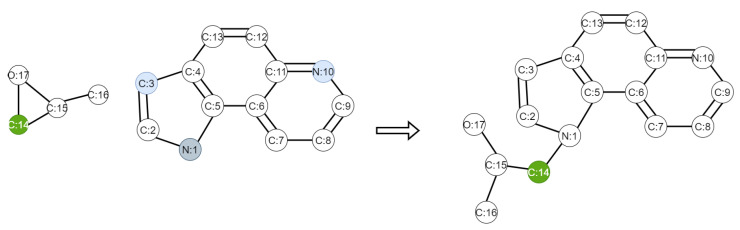
Propylene oxide reaction with pyrroloquinoline. The C, N, O in the figure represent chemical element symbols corresponding to Carbon atom, Nitrogen atom, and Oxygen atom, and the number followed denotes atom indices.

**Figure 4 molecules-27-05967-f004:**
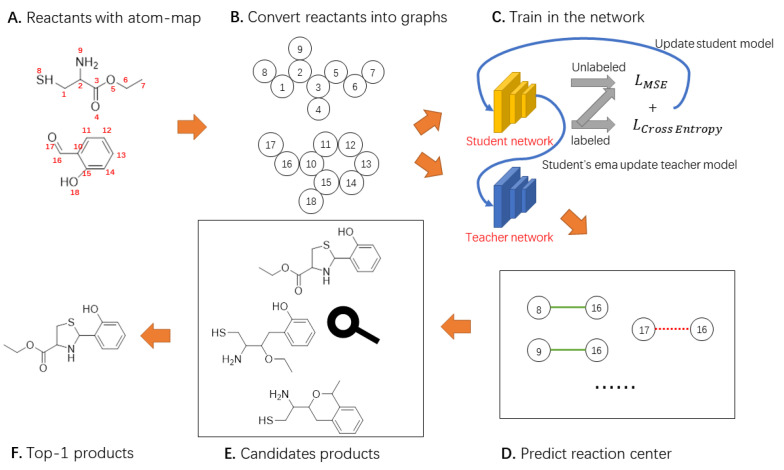
The general framework of our model. We describe the reactants as a graph. The student model learns reactant features to predict reaction centers. The teacher model uses the EMA of the student model to update parameters, then constructs better labels for the student to learn. Next, the candidate product set was constructed according to the reaction center. Finally, the score for each product is calculated.

**Figure 5 molecules-27-05967-f005:**
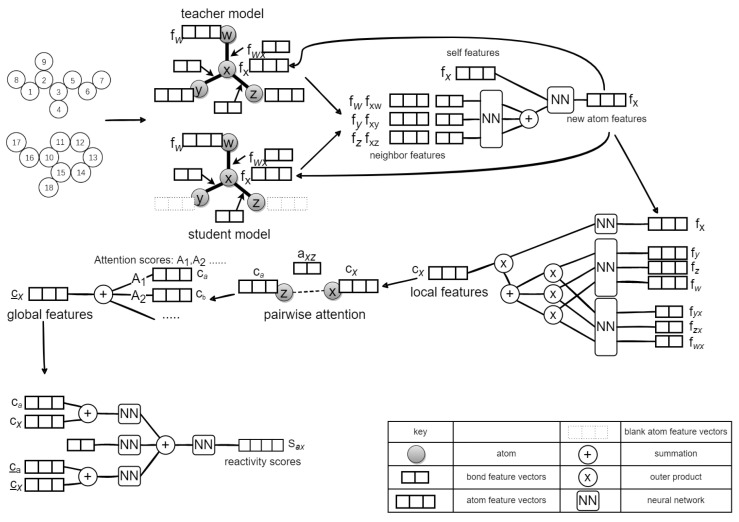
The workflow of the Mean Teacher Weisfeiler–Lehman Network (MT-WLN) for predicting reaction centers.

**Table 1 molecules-27-05967-t001:** Prediction accuracy of some methods. Top-*k* accuracy means that, if an algorithm proposes *k* potential products with the highest confidences, the probability that the corresponding ground-truth product is within these *k* products.

Method	Unlabeled Data Size	Top-1	Top-2	Top-3	Top-5
Seq2sep [14,15]	0k	80.3%	84.7%	86.2%	87.5%
WLN+WLDN [23]	0k	79.6%	-	87.7%	89.2%
WLN+I-WLDN [17]	0k	85.6%	90.5%	92.8%	93.4%
MT-WLN+I-WLDN	10k	85.7%	91.0%	92.7%	94.0%
MT-WLN+I-WLDN	20k	85.8%	91.1%	92.7%	94.1%
MT-WLN+I-WLDN	35k	**85.9%**	**91.1%**	92.8%	**94.1%**

## Data Availability

Not applicable.

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
