# Peer review of "Improving Chemical Reaction Prediction with Unlabeled Data"

_molecules, 2022, doi:10.3390/molecules27185967_

Round 1

Reviewer 1 Report

The article proposed a novel semi-supervised framework for template-free graph convolutional neural networks to predict chemical reactions. To overcome the intrinsic issue of data shortage in chemistry, the authors have extracted latent information in unlabeled dataset by Mean Teacher Weisfeiler-Lehman Network (MT-WLN) and improved the prediction accuracy of the reaction products. The manuscript gives a detail background of the research and tries to analyze the efficacy of the proposed methodology systematically. However, the performance improvement in “Table 1” seems to be not enough to be published as it is. Thus, I recommend a major revision to enhance the performance and quality of the contents as follows.

    The accuracy improvement of “MT-WLN+I-WLDN” compared with the “WLN+I-WLDN” is less than expected considering the inclusion of 35k unlabeled dataset. I recommend supplementing the contents in two ways. First, the change of the number of unlabeled data may have some effect on the performance. Second, there are no analysis and discussion about the accuracy of pseudo-labeled data from the teacher model. Analyzing the results would be beneficial to deep dive into the problem.

    More detail information seems to be required in the technical viewpoints: The criteria or the way of data selection from USPTO (400k) and Schneider (unlabeled training dataset; 35k). In the “Table 1”, Top-2 accuracy of WLN+WLDN is missing. In the manuscript, concerning the “Table 1”, some rough explanation of each approach (reference 10, 11, 13, 18) would be desirable for understanding of readers in terms of the difference, characteristics, or working mechanism.

    The sentence “Finally, we use Weisfeiler-Lehman Difference Network (WLDN) to rank candidate products, so that the main product has the highest score.” in the fourth page may have a mistake: “WLDN” à “I-WLDN”.

    The “b” of “Grb” in Eq. (1) & (2) should be modified into subscript.

    The full meaning of the LCEL (cross-entropy loss) and LMSE (the mean squared error) in Eq. (3) & (4) should be defined in the main paragraph.

Reviewer 2 Report

This manuscript, molecules-1889479-peer-review-v1- entitled "Improving Chemical Reaction Prediction with Unlabeled Data," is well written and has potential, but it should be more organized. In my opinion, a careful revision of the English language should be carried out as there currently are some unclear sentences. The study seems to be well designed. The methodology and results are technically sound. I recommend accepting this manuscript after revision. The main concerns are as follows:

1)     The title section should be edited and rewritten since it is too feneral and the authors did not mention the methodology

2)     The first paragraph should explain more about the importance of this study

3)     Improve the keywords by including only the phrases in the whole body.

4)     Some abbreviations in the paper have already not been addressed in the text like LHASA, SIMILES and USPTO.

5)     More recent references might support the first and second paragraphs of the introduction. References and literature are pretty old. There is no research reference in 2022 and one for 2021. The authors should read and use the newly published papers in their research.

6)     More literature review about the other methods is needed. The manuscript could be substantially improved by relying and citing more on recent literature about contemporary real-life case studies of sustainability and/or uncertainty, such as the followings.

·       Samani, S., Vadiati, M., Azizi, F., Zamani, E., & Kisi, O. (2022). Groundwater Level Simulation Using Soft Computing Methods with Emphasis on Major Meteorological Components. Water Resources Management, 1-21.

·       Meuwly, M. (2021). Machine learning for chemical reactions. Chemical Reviews121(16), 10218-10239.

7)     Providing a comprehensive flowchart is highly recommended by researchers, so please add a flowchart representing the methodology in the paper.

8)     USPTO data set is adopted as the case study. What are other feasible alternatives? What are the advantages of adopting this case study over others in this case? How will this affect the results? The authors should provide more details on this.

9)     Please provide all software used in this study.

10)  The Seq-Seq method should be better presented, with details of the models, etc.

11)  It is better to add more error criteria to better understand the model's ability.

12)  The discussion section in the present form is relatively weak and should be strengthened with more details and justifications.

13)  Comparison of the current study with previous research could be improved by more literature review.

14)  The limitations of the present study should be added to the paper, specifically for further research.

15)  It seems that conclusions are observations only, and the manuscript needs thorough checking for explanations given for results. The authors should interpret more precisely the results argument.

Reviewer 3 Report

Title: Improving Chemical Reaction Prediction with Unlabeled Data

Authors: Yu Xie, Yuyang Zhang, Chengbin Peng

It is an interesting report on models for prediction of chemical products. Some points should be clarified.

1. In the section Experiments the R(p) is assigned as the true reaction set and R(pp) as predicted set. Further, in the section Methods the R(p) is prediction set (??).

2. It is not clear what are Top-1-5 in Table 1.

3. Why Top-5 results are not presented in Figure 2?

4. There is practically no difference in predictions for training set, validation set and test set (Figure 1). The testing procedure should be explained and these results should be commented.

5. Reference [13] is wrongly cited in text.

Round 2

Reviewer 1 Report

The authors revised the manuscript and supplemented the results as recommended. Although the performance gain of the proposed model is not significantly greater than that of the "WLN+I-WLDN" method, the approach would be one of the important interests of readers. Therefore, I think the research is worth publishing as it is.

Reviewer 2 Report

Accept in present form